# On the low western Pacific sea levels observed prior to strong East Pacific El Niños

David J. Webb

National Oceanography Centre, Southampton SO14 3ZH, U.K.

**Correspondence:** D.J.Webb (djw@noc.ac.uk)

**Abstract.**

Research, based on both observations and ocean models, has indicated that strong East Pacific El Niños are preceded by the development of unusually low sea levels at the western end of the North Equatorial Trough early in the year. This results in an increase in the strength of the North Equatorial Counter Current, which aided by low sea levels due to the annual Rossby wave, then transports West Pacific Warm Pool water to the central and eastern Pacific.

Here an ocean model is used to investigate the factors affecting sea level prior to the 1982-1983 East Pacific El Niño. The results indicate that during this period the reduction in sea level was caused by Ekman pumping, due to the local winds, acting over a period of many months. The north-south distances involved mean that such upwelling can be connected to the westerly wind phrase of Madden-Julian Oscillations.

Since the 1980s the quality and quantity of global wind measurement has steadily improved. So, if the Ekman pumping hypothesis is correct, the better quality wind data available prior to the 1997-1998 and 2015-2016 East Pacific El Niños should also show unusually large Ekman pumping in the same region, over a similar long period. This is tested and found to be correct.

However, a study of the integrated Ekman pumping for the period 1976 to 2015, indicates that in some years similar events did not develop despite a comparable amount of pumping during the first part of the year. The results imply that significant Ekman pumping early in the year is a necessary but not sufficient requirement for the development of a strong East Pacific El Niño.

## 1 Introduction

The North Equatorial Countercurrent (NECC) flows from west to east across the Pacific at latitudes between 5°N and 10°N. The total transport is in the range 10-30 Sv, making it comparable with the major northern hemisphere currents (Knauss, 1961; Wyrtki, 1974a,b; Johnson et al., 2002).

Physically it is a geostrophic current lying between the sea level minimum of the North Equatorial Trough and the sea level maximum of the North Equatorial Ridge (Sverdrup, 1947; Neumann and Pierson, 1966). However it is also a shallow current with a depth of only two to three hundred metres, the north-south pressure differences near the surface being balanced by large north-south gradients in the near surface density surfaces (Wyrtki and Kilonsly, 1984; Taft and Kovala, 1981, 1982; Bryden and Brady, 1985).

The current is important because it carries warm water eastwards out of the West Pacific Warm Pool (Johnson et al., 2002). It is also unusual in that it flows in the opposite direction to the steady trade winds of the North Pacific. Montgomery and Palmen (1940) originally suggested that the current flowed downhill from west to east across the Pacific but, using geostrophic theory, Sverdrup (1947) showed that it was the result of the shear in the wind field at latitudes near the Inter-Tropical Convergence Zone (ITCZ) (See also Munk, 1950). The current also lies in a region where the inverse of the Coriolis term is a rapidly changing function of latitude. As a result small changes in the position of the trough can have a significant effect on the velocity of the current.

Early studies of the current were limited by lack of data but, using data from bathythermographs, Meyers (1979) showed that the depth of the North Equatorial Trough had a significant annual signal, which he identified as being due to an annual Rossby wave.

Further developments in ocean instrumentation have provided more information on the NECC, satellite observations of sea surface temperature (SST) showing tropical instability eddies (often called waves: Hansen and Paul, 1984; Chelton et al., 2000; Kennan and Lament, 2000) that both warm the Equatorial Current and cool the NECC (Menkes et al., 2006; Jochum et al., 2007), the effect possibly being least in El Niño years (Yu and Liu, 2003).

Satellite altimeter measurements of sea level have errors of only a few centimetres and, as the North Equatorial Trough has a depth of order one metre, this allows the annual and interannual changes of both the trough and the NECC to be studied in detail (Zhao et al., 2013; Tan and Zhou, 2018).

## 1.1 Connection with the El Niño

Originally El Niño was the name of a southward flowing current off the coast of Peru which affected fisheries (Wyrtki et al., 1976; Philander, 1989). However the studies reported in Love (1972, 1975) and Wyrtki et al. (1976) connected it to the very large changes in surface temperature and thermocline depth observed in the nearby East Pacific Cold Pool.

Wyrtki (1973) was the first to make the further connection between this oceanographic El Niño and the strength of the NECC in the western Pacific. He also suggested that the NECC might trigger an El Niño by transporting West Pacific Warm Pool water to the east.

Meyers and Donguy (1984) then showed that the NECC transport increased by between 25% and 50% while the strong 1982-83 El Niño was developing. They also found that the total transport of warm water (>28°C) by the NECC was consistent with their estimate of the loss of warm water from the West Pacific Warm Pool.

Although Wyrtki's hypothesis was not developed further, his work complimented the study by Bjerknes (1969) which showed a correlation between water temperatures in the equatorial Pacific and the Southern Oscillation (Philander, 1985). This led to

further uses of the term El Niño to describe both its world wide influence and the air-sea interaction (or mechanism) behind the event (Cane, 2011).

Bjerknes (1969) also argued that the east-west temperature gradient along the Equator was involved and this is usually taken to imply that the temperature of the East Pacific Cold Pool is a key part of the El Niño mechanism. However Clarke (2014) doubts this, because the east Pacific winds are only weakly affected by an El Niño. If he is correct, temperature changes in the Cold Pool are a consequence of and not a cause of El Niños.

Further studies showed that Bjerknes' correlation was strongest when using ocean temperatures from the central Pacific. As
a result most modern studies of the Southern Oscillation use the Nino 3.4 index to measure the strength of an El Niño (Larkin and Harrison, 2005). This is based on the mean sea surface temperature anomaly in the central Pacific between 170°W and 120°W, and between 5°S and 5°N.

As well as affecting surface winds in the western and central Pacific, the atmospheric El Niño requires the centre of deep atmospheric convection to move from the maritime continent and the western Pacific into the central Pacific. Evans and Webster
(2014), following Gadgil et al. (1984) and Zhang (1993), confirmed that this requires sea surface temperatures of 28°C and above during wet seasons and even higher temperatures during dry periods.

In the central Pacific, such temperatures are found on the Equator during an El Niño, but eastern Pacific equatorial temperatures never reach this value (Philander, 1985, Fig. 2). Instead the eastward limit of warm water on the Equator is closely correlated with the Southern Oscillation Index (Picaut et al., 1996).

In the eastern Pacific, temperatures sufficient to trigger deep atmospheric convection are often found further north in the ocean off Central America and along the path of the NECC close to the ITCZ.

Meteorological studies also showed that the start of an El Niño was often connected with one or more Madden-Julian Oscillations (MJOs) (Madden and Julian, 1971, 1972). In his review, Zhang (2005) describes MJOs as individual events with lifetimes of between 30 and 100 days and with the property that there is never more than one major event in existence at a
time. Each one consists of an area, spanning the Equator, in which there is an increased density of short lived convective cloud groups.

The events progress eastwards at speeds around $5\ \mathrm{ms}^{-1}$, starting in the western Indian Ocean and dying out after passing the western Pacific. At sea level they are associated with inflowing winds from the east and west, but in the Pacific the westerly winds may extend past the centre of the cloud groups.

## 1.2   El Niño models and theories

Following Wyrtki and Bjerknes, most theoretical studies of the ocean's contribution to El Niño have focussed on the Cold Pool. This is a narrow band of low sea surface temperatures, centred on the Equator in the eastern Pacific. During a normal year this results from the wind generated upwelling of cold water from depths of around 200m, but during an El Niño the surface thermocline becomes thicker and less cold water is upwelled.

Early theoretical studies, using analytic models or numerical models with a few layers (Hurlburt et al., 1976; McCreary, 1976, 1985; McPhaden, 1981, 1993), showed that the changes in the Cold Pool could be generated by wind variations in the

western and central Pacific. This occurred via the propagation of baroclinic Kelvin waves in the equatorial wave guide. The studies also showed that only winds close to the Equator were responsible (McCreary, 1976) and that off equatorial currents, such as the NECC, were not connected with the El Niño changes to the Cold Pool.

With the development of computer power, it was eventually possible to run ocean models with much more detailed physics and with enough horizontal and vertical resolution to resolve the key currents and structure of the ocean (Bryan, 1969; Semtner, 1974; Philander, 1985; Cox, 1989; Smith et al., 1992). The versions developed by Cox (1989), Madec et al. (1998) and Griffies et al. (2005) have been widely used in stand-alone mode and, coupled to an atmospheric model, for studies of climate.

Unfortunately the NECC can be poorly represented in such models (Lengaigne et al., 2002; Sun et al., 2019). For ocean only
models, Harrison et al. (1990) found that the poor results were largely due to errors in the atmospheric forcing datasets.

Yu et al. (2000) concentrated on the forcing of the NECC and showed that a weak NECC resulted from the atmospheric reanalysis datasets underestimating the curl of the wind field near the latitudes of the ITCZ (Byrne et al., 2018) and overestimating the strength of easterly winds on the Equator.

The study also found that, of the datasets studied, the ECMWF reanalysis was best at generating a realistic NECC. This may
be because the ECMWF model was using a spherical harmonic expansion which requires less smoothing than normal finite difference schemes to overcome non-linear instabilities.

There is a related problem with coupled ocean-atmosphere models which fail to generate realistic El Niños (Guilyardi et al., 2009; Ham and Kug, 2012; Flato et al., 2013; Hsu et al., 2021). This again may be due to the resolution of the atmospheric component being insufficient to accurately represent features such as the ITCZ.

Further studies using observational data have indicated that there are two types of El Niño (Larkin and Harrison, 2005; Ashok et al., 2007; Tan and Zhou, 2018). The most common are the central Pacific (CP) or Modoki events, in which the centre of atmospheric convection moves to the central Pacific. More rarely the centre of convection moves even further east to give the strong eastern Pacific (EP) El Niños (Cai et al., 2014). These behave like the classic El Niños (Philander, 1985), which Wyrtki (1974a) proposed were triggered by the NECC.

**1.3  Wyrtki's hypothesis**

Although neglected for many years, Wyrtki's hypothesis is supported by more recent studies using altimeter data (Zhang and Busalacchi, 1999; Hsin and Qiu, 2012; Zhao et al., 2013; Wijaya and Hisaki, 2021).

These confirm the increased transport by the NECC during EP El Niños, Wijaya and Hisaki (2021) showing that increases in the transport of the NECC in the western Pacific occur around three months before the corresponding change in the Nino 3.4
index. Also in the western Pacific, Zhao et al. (2013) show that there is a delay of between 40 and 90 days between changes in the depth of the North Equatorial Trough, north of the NECC, and changes in sea level near the Equator.

The results thus leave open the possibility that the increased transport has a role in triggering subsequent EP El Niños, rather than being part of the ocean's response to the event. The studies also indicate similar changes in the NECC during CP El Niños, indicating that both types of El Niño may be affected by the NECC in the western Pacific

Further support for the hypothesis comes from Webb (2018) who used results from a high resolution global version of the Nemo ocean model to study the strong EP El Niños of 1982-1983 and 1997-1998.

For this study Nemo was forced by the Drakkar dataset (Dussin et al., 2016), based on the ECMWF reanalysis fields, and produced sea level and sea surface temperature fields in the equatorial Pacific in close agreement with satellite observations (Webb et al., 2020). The results indicate that the modified ECMWF reanalysis fields accurately represented the wind stress curl
and so generated a realistic NECC.

Unlike previous studies, Webb (2018) concentrated on water with temperatures above 28°C, sufficient to trigger deep atmospheric convection. The study showed that, while the strong El Niños were developing, the NECC carried warm water into the eastern Pacific far in advance of water with similar temperatures on the Equator.

The study also helped explain why the NECC does not carry Warm Pool water into the eastern Pacific every year, by showing
that in normal years tropical instability waves mix in cold upwelled water from near the Equator and that this rapidly reduces temperatures in the NECC (See: Baturin and Niiler, 1997). In El Niño years the eddies are weaker so their effect on the NECC is reduced.

It also found that the start time of strong El Niño events resulted in warm water reaching the central and eastern Pacific at a time of year when the annual Rossby wave meant that the NECC was strongest there. This resulted in the warm water arriving
off South America around the end of the year, thus providing a physical mechanism to explain the name originally given to classic EP El Niños.

Finally the study showed that both the strong 1982-83 and 1997-98 El Niño events started when the depth of the western section of the North Equatorial Trough was unusually low. As a result the NECC transport, and the flux of Warm Pool water eastwards, were both much stronger than usual.

## 1.4  Low sea levels in the western Pacific

Webb (2018) suggested that the low sea level in the trough was due to an unusually strong annual Rossby wave prior to each El Niño. However further investigation showed that this is unlikely, as in each case the wave is not unusually large as it crosses the central Pacific.

Another possible explanation is that the stratification in the Pacific, prior to an El Niño, is sufficient to focus the Rossby
wave and so cause lower than normal sea levels. Alternatively the low sea level may not be due to the Rossby wave but instead is due to some other process associated with the wind field. This could be a local feature occurring at or just before the period when sea level drops. It could also be a response to forcing elsewhere in the ocean, which later propagates into the western Pacific.

To help clarify the cause of the drop in sea level, this paper starts by reporting on series of short ocean model runs which
focus on changes in the western Pacific prior to the 1982-1983 El Niño. As in Webb (2018) this allows the development of hypotheses, which are then tested by treating them as predictions for the strong El Niños of 1997-1998 and 2015-2016 for which better observationals are available.

The global ocean model used for the tests is an updated version of Occam (Webb et al., 1997; de Cuevas et al., 1999). In each run, the model is initialised from one of the archive datasets from the original run of the Nemo model and forced with ocean surface stresses calculated during the same run.

The results imply that the low sea levels, at the western end of the North Equatorial Trough, are due to local Ekman pumping acting over a period of many months.

This hypothesis is tested against the data from the original run of the Nemo 1/12° global ocean model, initially during the growth of the 1982-83 El Niño and then for the strong EP El Niños of 1997-98 and 20015-16. A similar comparison covering the whole period from 1976 to 2015 is also carried out.

## 1.5   Structure of the report

In the first part of the paper, section 2 describes the model being used and how the model fields and forcing were converted from the 1/12° Nemo grid to the 1/4° Occam grid.

Section 3 reports on tests carried out to validate the lower resolution model. In these the model was started from the Nemo archive datasets from early January 1981 and 1982 and then run for a year using the surface wind stresses from the same year.

The following sections report tests with different wind forcing and initial conditions. This includes runs where the model was again started from early January 1981 and 1982, but with the wind forcing from the opposite year. The results show that the main changes in sea level depend primarily on the wind field and, to first order, are independent of the stratification at the start of the run.

The tests are repeated but starting later in the year when the annual Rossby wave has developed and is starting to cross the central Pacific. This is to check whether it is responsible for the sea level drop in the western Pacific. The results are similar to the earlier test, indicating that the wind generated Rossby wave is not responsible.

A final test is designed to investigate whether the winds causing the drop in sea level are local to the western equatorial Pacific or propagate in from other parts of the ocean. The results indicate that it is the local winds are responsible.

The most likely way in that the local winds could affect sea level in the North Equatorial Trough is through Ekman pumping and the resulting rise in density surfaces within the ocean. In the second part of the paper this hypothesis is tested.

Section 7 compares the expected uplift due to Ekman pumping during 1982 with the rise in density surface in the Nemo 1/12° model. The results are promising so in section 8, similar tests are also carried out for the 1997-98 and 2015-16 EP El Niños.

Finally a similar survey of the whole period 1976 to 2015 is carried out to see if strong Ekman pumping in the western section of the North Equatorial Trough during the early part of the year is common, and whether it might also be a sufficient condition for the development strong EP El Niños.

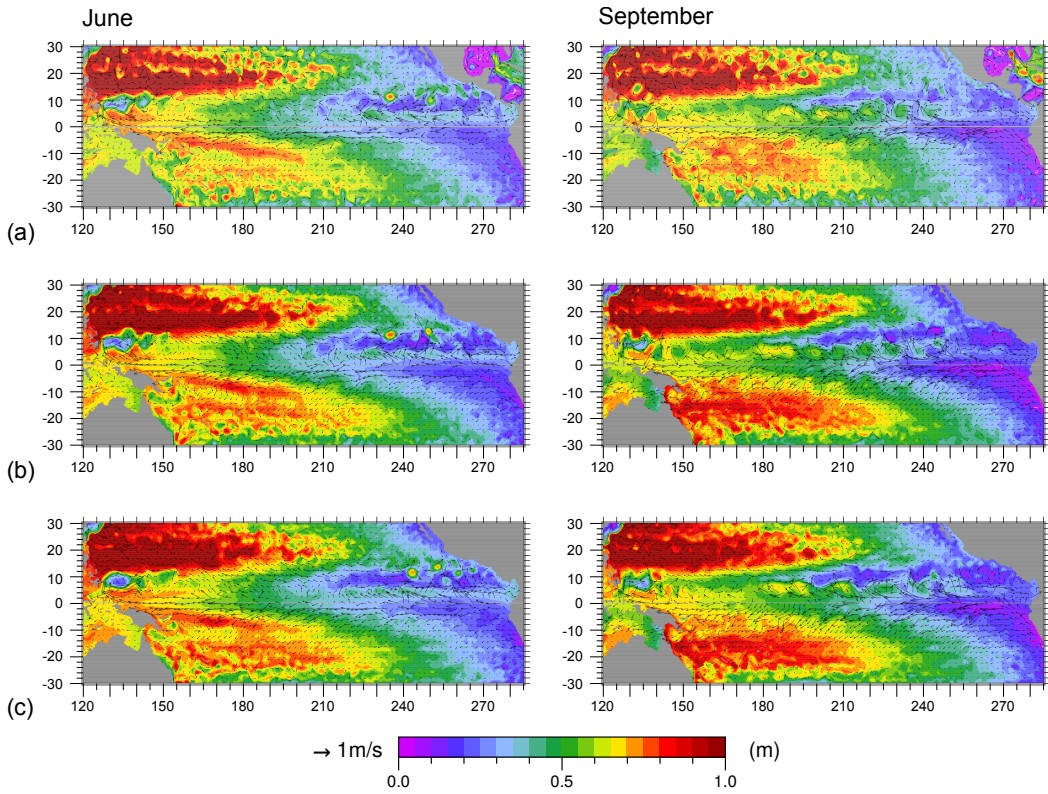

**Figure 1.** Sea level (m) plus surface currents on the 6th June and 9th September for (a) Nemo in 1981, (b) Occam in 1981, started January 1981 (c) Occam in 1982, started January 1982 but with 1981 winds.

## 2  The Occam 1/4° global ocean model

Occam is a primitive equation model, based on the Bryan-Cox-Semtner series of models (Bryan, 1969; Cox, 1989; Semtner, 1974). It uses a regular latitude-longitude grid for all the oceans except the North Atlantic and Arctic. For the latter basins it uses a second rotated latitude-longitude grid which is matched to the first grid at the Equator. The Bering Strait between the Arctic and the North Pacific is modelled with a simple channel model.

Occam uses 1/4° resolution in both longitude and latitude. In the vertical it has 66 levels, instead of the 75 levels of the Nemo run, and makes use of an existing global topography that was checked against a database of critical oceanographic sills (Thompson, 1996). Both the Occam and Nemo models have increased vertical resolution in the surface layers, Occam using 24 layers in the top 300m and Nemo 34 layers, the difference being primarily due to Nemo's extra resolution in the top 100m.

Occam uses harmonic mixing in the horizontal and the scheme of Pacanowski and Philander (1981) for vertical mixing. For horizontal advection it uses the second order split-quick scheme (Webb et al., 1998) for both momentum and tracers. For the vertical advection of momentum it uses the scheme of Webb (1995).

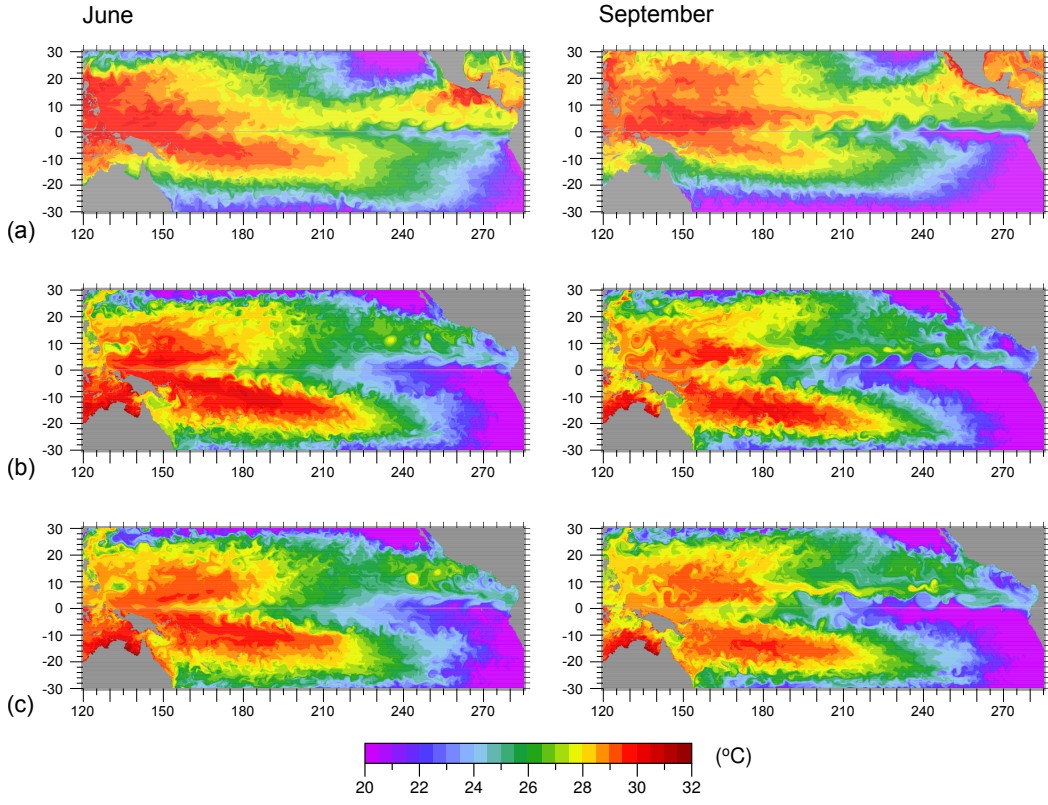

**Figure 2.** Surface temperature (°C) on the 6th June and 9th September for (a) Nemo in 1981, (b) Occam in 1981, started January 1981 (c) Occam in 1982, started January 1982 but with 1981 winds.

Occam was chosen for these tests primarily because of its computational efficiency and because the amount of computer time available was limited. The efficiency arises partly because, unlike Nemo, it uses a regular grid and partly because the code includes fewer of the complex physical options included in Nemo.

However most ocean models are limited not by the speed of the processor but by the length of time needed to transfer data from main memory. Occam's main advantage is that it overcomes this by vectorising the code in the vertical[1].

## 2.1 Initialisation and forcing

The model runs reported here were initialised by averaging the archived data from the high-resolution Nemo model onto the Occam grid. Variable values within each Occam ocean cell were calculated by averaging over the intersection of the Occam and Nemo cells. In the case of vector quantities, vectors were rotated to the Occam grid before averaging.

---

[1]This means that all the variables needed to time step the cells in a vertical column are held in high speed cache at the same time and no extra references to main memory are required. Also when moving from one column of ocean cells to the next, the cache already contains most of the required data.

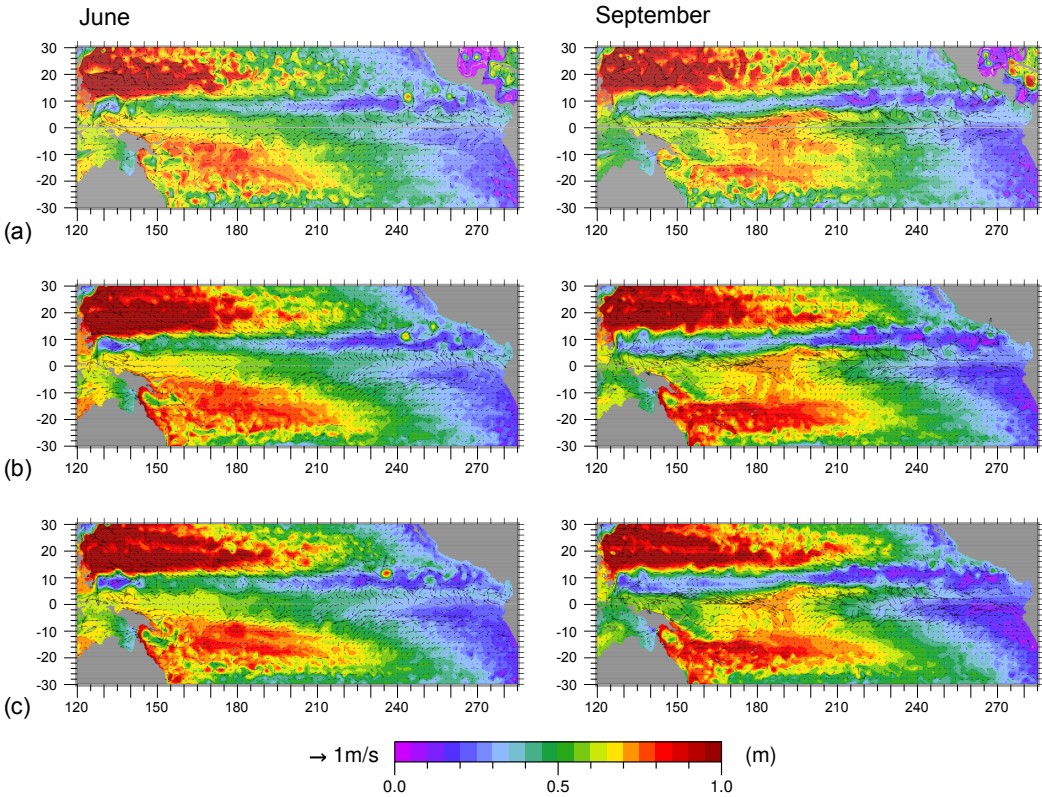

**Figure 3.** Sea level (m) plus surface currents on the 6th June and 9th September for (a) Nemo in 1982, (b) Occam in 1982, started January 1982 (c) Occam in 1981, started January 1981 but with 1982 winds.

The runs were also carried out with zero flux of heat and salt across the ocean surface. Each run is for only a few months
and, although the surface temperatures and salinities are affected, this approximation allows the analysis to concentrate on the effects of wind stress, and of advection and diffusion within the ocean.

Each Nemo archive datasets contain the ocean and forcing fields averaged over the previous 5 days. Thus when initialising the Occam model and when specifying the wind forcing, the time of each Nemo dataset is set to the central time of the averaging period. However when specifying particular Nemo archive files, the original archive date is used.

The use of five day averages filters out higher frequency components of the wind field and also means that the ocean state used to initialise the Occam model may be unusually smooth. The resulting lack of high frequency oscillations may affect vertical mixing in the ocean but should otherwise have little effect on lower frequency variations in sea level and current velocity, which are the focus of this study.

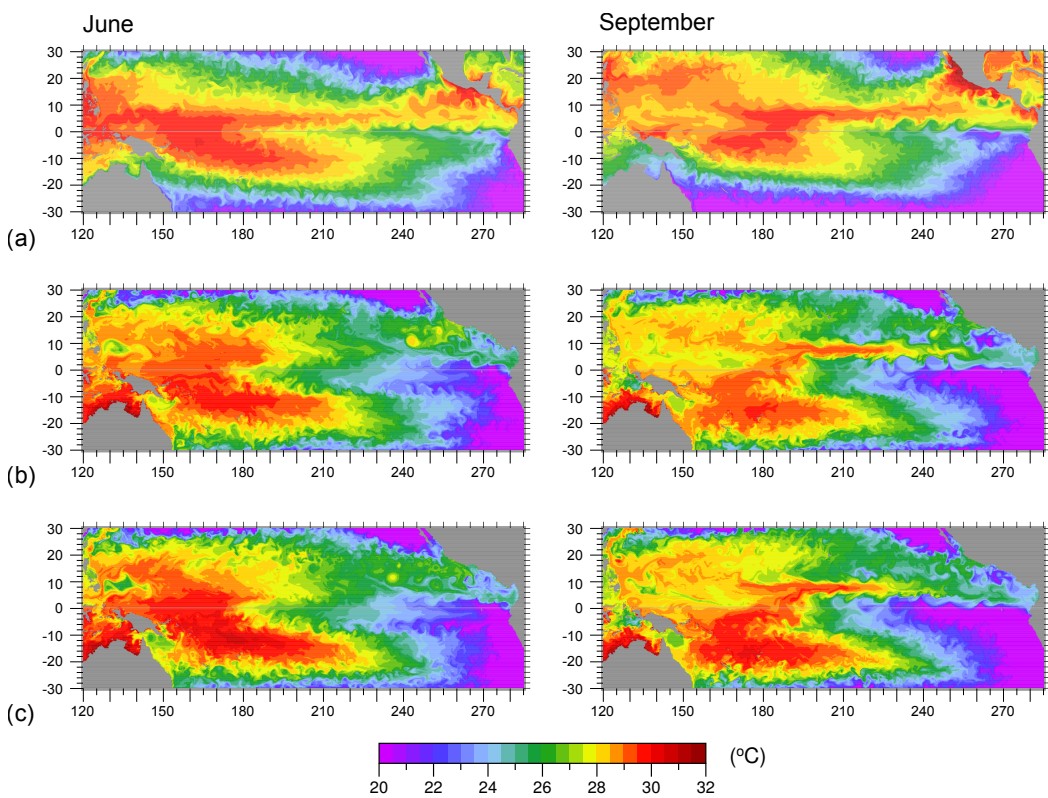

**Figure 4.** Surface temperature (°C) on the 6th June and 9th September for (a) Nemo in 1982, (b) Occam in 1982, started January 1982 (c) Occam in 1981, started January 1981 but with 1982 winds.

## 3  Validation

Previously both the Nemo and Occam models have been widely used for successful oceanographic research, but here the use of a lower resolution Occam model to investigate features seen in the higher resolution Nemo run is validated by comparing model results in two runs. Each run was over a full year and each used the Nemo archived surface wind stresses from that year.

In the first run, designed to compare performance in a normal year, Occam was initialised from the Nemo archive datasets from the 5th January 1981. In the second, a test of a strong El Niño year, the model was started from the 5th January 1982.

The use of 1981 as a representing a normal year has the advantage that, there being no other known special considerations that might affect the results, to a first approximation developments in 1981 should reflect what would have occurred in 1982 had there been no El Niño.

## 3.1   Comparison of 1981 results

In Fig. 1, the top two pairs of figures show sea level from the Nemo and Occam models on the 6th June and the 9th September
1981. The 6th June is chosen because the corresponding date in 1982 can be used to check the early deepening of the North
Equatorial Trough in the western Pacific that occurs prior to the 1982-1983 El Niño. Similarly the 9th September is chosen as
a check on how well Occam reproduces the increased depth of the trough, and corresponding increased strength of the NECC,
across the whole of the Pacific, during the autumn months.

At the largest scales the Occam model is in close agreement with Nemo, the main differences arising from slightly more
extreme maxima and maxima. Thus in September, maximum sea level in the South Pacific Gyre is slightly higher in Occam.
Similarly the minimum sea levels in the Cold Pool on the Equator and within the North Equatorial Trough are slightly lower.

This paper is primarily concerned with the behaviour of the NECC and the associated sea levels in the North Equatorial
Trough. Here Occam shows a similar westward propagation of the trough during the year and a similar growth of the ridge
on the southern side of the NECC. The September figures also show a similar development in Occam of the short wavelength
features on the ridge, associated with the growth of tropical instability eddies.

In June, both Nemo and Occam show that the trough is particularly weak in the central Pacific resulting in an almost non-
existent NECC. However in September, with a greater contrast between ridge and trough, both models show a stronger NECC.

At the western ends of the trough and ridge, Occam shows a similar development of the high and low sea level regions
associated with both the Halmahera and Mindanau eddies (Kashino et al., 2003) and the initial meanders of the NECC. In
September these decay in amplitude, in contrast to the growth of the trough and ridge features seen in the central and eastern
Pacific.

The development of sea level during the year at 6°N in both Nemo and Occam is shown in panels (a) and (c) of Fig. 5.

The average slope in sea level in Occam matches that of Nemo and in the central and eastern Pacific, Occam shows a similar
development of the ridges due to the growth and propagation of tropical instability eddies. The figure also shows the low sea
levels due to the propagation on the annual Rossby wave at 6°N. This starts in the eastern Pacific early in the year and reaches
the dateline in July and August.

The corresponding sea levels on the Equator are shown in panels (a) and (c) of Fig. 6. Again the overall east-west slope is
similar and Occam reproduces the eastward propagating equatorial waves that can be generated by westerly wind bursts in the
western equatorial Pacific. There are differences between the two model runs but in both models sea level is a maximum either
close to the western boundary or near 150°E.

Sea surface temperatures, in June and September 1981, are shown in the top pairs of panels in Fig. 2. As Occam includes no
surface warming or cooling it is expected to show no seasonal changes and this is seen to be the case, Occam showing warmer
temperatures in the southern hemisphere in both June and September and cooler temperatures in the north.

The differences are largest off Central America, where Occam fails to reproduce the East Pacific Warm Pool, and in the
western section of the South Pacific, where there is little reduction of Occam temperatures between June and September. The
regions of cold water, with temperatures at or below 20°C, also behave very differently in Occam and Nemo.

Along the line of the NECC at 6°N, Occam fails to show the warming during the first half of the year, but by September it does show a narrow band of warm water being advected into the eastern Pacific by the NECC. It also shows it being eroded by tropical instability eddies.

## 3.2 Comparison of 1982 results

Corresponding sets of panels for 1982, during the development of the 1982-1983 El Niño, are shown in Figs. 3, 4, 5 and 6.

In Fig. 3, Occam shows, in agreement with Nemo, a much deeper North Equatorial Trough in early June together with a well developed NECC. This is emphasised further in September with a deep trough in the west which extends almost as far as the Mindanao Eddy.

The Hovmöller diagrams of Fig. 5 show that in 1982 the annual Rossby wave, in both Occam and Nemo, links with the region of low sea level that develops at the western end of the trough. Webb (2018) argued that it was this drop which generated the increase the NECC strength which triggered the development of a classic oceanographic El Niño.

At the Equator (Fig. 6), Occam shows, in agreement with Nemo, the movement of the maximum sea level away from the western boundary to a region near 200°E. In Figs. 3 and 4 this is seen to correspond to the region of high sea level on the Equator where the warmest Equatorial temperatures are also found.

## 3.3 Summary of comparisons

The results show that the Occam model is capturing the key features in the sea level field which affect the strength of the NECC, the strength of tropical instability eddies and processes occurring on the Equator. It is less effective at capturing changes to sea surface temperature field. This may partly be due to the lack of surface heat fluxes in this version of the model. It may also be partly due to the much reduced horizontal resolution of the model.

However by bearing these strengths and weaknesses into account, there is no reason why it cannot be used to study the effect of the winds and the initial state of the ocean on the development of the 1982-1983 El Niño, as is done here.

## 4 Tests using wind stresses from different years

Although Webb (2018) identified a number of processes contributing to the development of a strong El Niño in 1982 and 1997, the study was not able to explain why the events started in those years and not, say, a year earlier or a year later.

One possibility, suggested by Webb (2018), is that there was something different about the stratification of the ocean in early 1982 and 1997, which focused the annual Rossby waves, so that they had a larger amplitude than usual once they reached the western Pacific. Another possibility is that the difference was due to the wind. This may have generated a stronger than normal annual Rossby wave that year. Alternatively some independent wind event may have occurred which lowered sea level in the western Pacific around the time that the leading edge of the annual Rossby wave arrived.

In an attempt to distinguish between these possibilities, the two Occam runs starting in January 1981 and 1982 were repeated, but this time forced by the winds from the opposite year. If focusing is important then the January 1982 ocean, forced by the

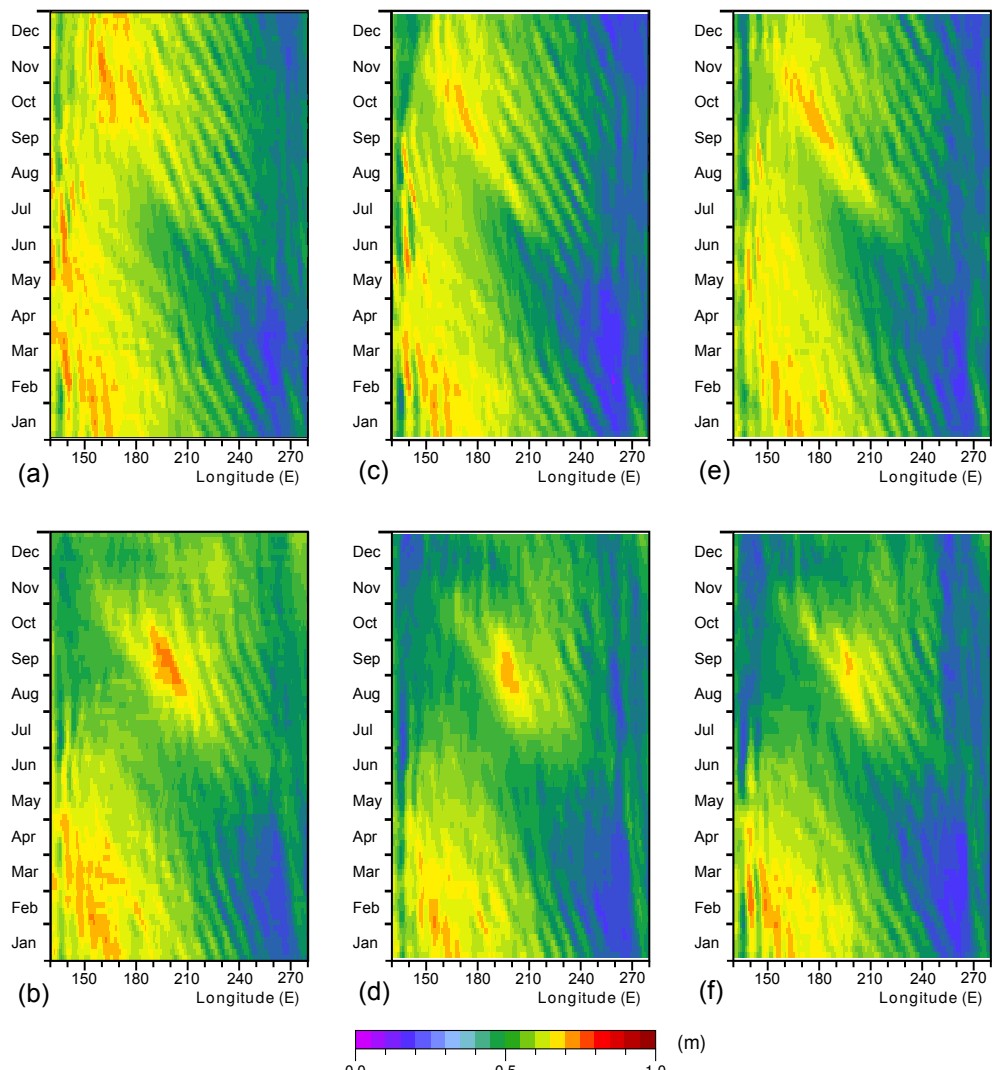

**Figure 5.** Hovmöller diagrams of sea level (m) at 6°N in the Pacific for (a) Nemo model in 1981, (b) Nemo in 1982, (c) Occam starting January 1981, (d) Occam starting January 1982, (e) Occam starting January 1982 but with 1981 winds, (f) Occam starting January 1981 but with 1982 winds. The figures are based on 1° averages of the model data.

1981 winds, might generate a similar enhanced annual Rossby wave in the western Pacific. Alternatively if the winds are important then the January 1981 run, forced by the 1982 winds, might generate the low sea level in the western Pacific.

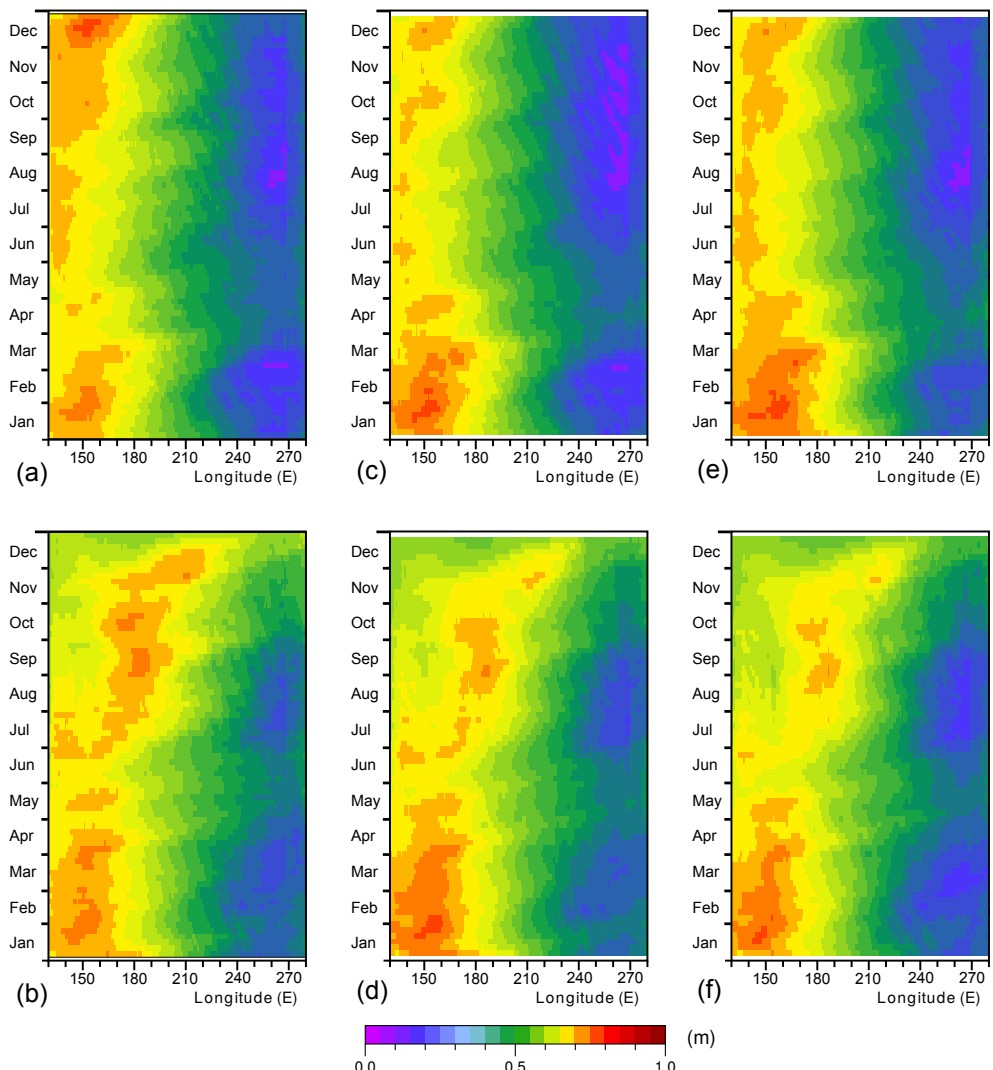

**Figure 6.** Hovmöller diagrams of sea level (m) on the Equator for (a) Nemo model in 1981, (b) Nemo in 1982, (c) Occam starting January 1981, (d) Occam starting January 1982, (e) Occam starting January 1982 but with 1981 winds, (f) Occam starting January 1981 but with 1982 winds. The figures are based on 1° averages of the model data.

## 4.1   January 1982 ocean forced by 1981 winds

In this test, the model was initialised from the Nemo archive dataset dated the 5th January 1982, but then forced with winds from 1981. Figure 5e shows how sea level developed at 6°N. During the first few months of the year the annual Rossby wave develops and propagates westwards as normal but in mid-ocean the amplitude declines so by the time the wave reaches 180°E the signal is weak.

On the Equator (Fig. 6e), sea level starts high in the western Pacific, as in the earlier run with 1982 winds forcing the 1982 ocean (Fig. 6d), but as the year develops it stays in the west and there is no movement into the central Pacific.

Figures 1c and 2c show the sea level and temperature fields for June and September. Previous runs with 1982 winds show that by June the depth of the trough is starting to increase more than in a normal year but here this is not happening and instead the response is closer to that of Nemo in 1981. Similarly in September the trough fails to develop further and the response again lies closer to that of Nemo in 1981.

The surface temperature field is also closer to the Nemo results from 1981, with the bulk of the Warm Pool water remaining in the west. However there is a thin core of warmer NECC water extending into the eastern Pacific, which is not present in the run started in January 1981 with 1981 winds. This indicates that there was some change in the structure of the ocean between January 1981 and 1982 which aided the transport of warm water eastwards by the NECC.

Despite this, the main conclusion from this run is that the ocean state at the start of 1982 was not sufficient to trigger an El Niño in mid-year. Although a reasonable annual Rossby wave was generated early in the year, this was not focused and did not generate or contribute to the lowering of sea level in the western Pacific.

## 4.2 January 1981 ocean forced by 1982 winds

In the second test, the ocean is initialised from the Nemo archived dataset dated the 5th January 1981, but then forced with 1982 winds. Figure 5f shows the sea level during the year at 6°N. The annual Rossby starts as before, but this time it continues past 180°E and links up with a region of low sea level in the western Pacific - as might be expected at the start of a strong El Niño.

On the Equator (Fig. 6f) the ocean again starts with high sea levels in the west. In mid-year these move into the central Pacific - again as might be expected at the start of a strong El Niño.

Figures 3c and 4c show that in mid-year the distribution of sea surface temperature is similar to the other Occam runs, but sea level shows a well developed North Equatorial Trough with a strong NECC developing on its southern slope.

By September the trough has developed further and the temperature plot shows warm surface water being advected rapidly into the eastern Pacific. The area of warm water involved is not as large as in the original Nemo run in 1982, but as with Fig. 4b, this is probably a result of setting the surface flux of heat to zero in these test runs.

One conclusion from this test is that the ocean will produce an El Niño like response when forced by the winds from an El Niño year. This has been reported before, but based primarily on the large sea level and temperature anomalies that develop in the eastern Pacific Cold Pool towards the end of the first year of a classic oceanographic El Niño.

The present study is different in that it shows the winds having a significant effect during the first half of the year, well before any changes in the Cold Pool region are noticeable. The results also show that by mid-year the 1982 winds have produced significant changes in the western section of the North Equatorial Trough. What is not clear is whether the increased depth of the trough is due to the mean winds during the first part of the year or due to one or more isolated events.

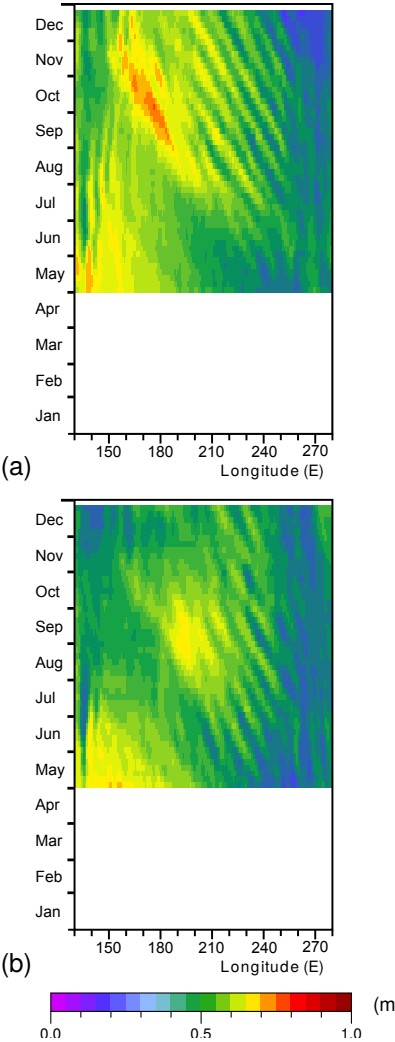

**Figure 7.** Hovmöller diagram of sea level (m) at 6°N during 1982 for (a) when started from the Nemo model archive from the 30th April 1982 but forced by 1981 winds, (b) started from the Nemo model archive from the 30th April 1981 but forced by 1982 winds.

It is also not clear whether it is due to a local change in the winds along the line of the trough, or whether the key wind forcing occurs elsewhere and the signal propagates into the region lowering the level of the trough. The remaining model runs reported here attempt to obtain a clearer answer to these questions.

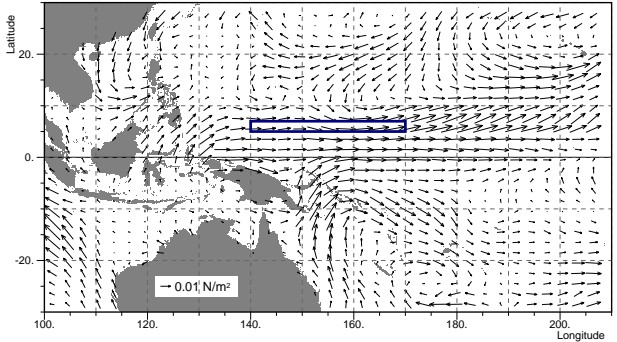

**Figure 8.** Mean wind stress vector anomaly ($\mathrm{Nm^{-2}}$), for the period 16th March to the 8th August 1982 relative to the same period in 1981. The blue rectangle shows the region between 140°E and 170°E and between 5°N and 7°N.

## 5   Tests starting in late April

One possibility that has not been discounted is that, early in 1982, the winds generated a Rossby wave, or similar, which was later responsible for the sea level drop in the western Pacific. To test this hypothesis, the model was started from the Nemo archive dataset dated the 30th April 1982 and then forced with the 1981 winds. The start date is just before the start of the drop in sea level in the western Pacific. If the Rossby or other waves are responsible, then by that date they should be established enough to reproduce the drop in sea level despite the change in the wind field.

Fig. 7a shows how sea level develops along 6°N. The annual Rossby wave starts propagating across the central Pacific as normal but near the dateline its amplitude is greatly reduced and there is no connection with the lower sea levels of the far western Pacific. Thus, although the test does not exclude events occurring before the 30th April having some impact, they cannot be the prime cause of the sea level drop that increases the strength of the NECC near 160°E.

The complimentary test was also carried out, in which the ocean was started from the 30th April 1981 but forced with 1982 winds. The result, shown in Fig. 7b, shows that sea level does drop. This implies that it is the winds after the 30th April 1982 that are responsible.

## 6   The role of local winds in the western Pacific

Figure 8 shows the difference between the 1982 and 1981 wind stress vectors for the Pacific when averaged between the 16th March and the 8th August. It shows that in the western Pacific near the Equator, there is a significant westerly component to the wind stress anomaly. North of New Guinea this drops to zero near 10°N. This distance is typical of the atmospheric equatorial Rossby radius, a scale which also determines the northward extent of Madden-Julian Oscillations (Madden and Julian, 1971, 2012).

MJOs are reported to be stronger over the western Pacific prior to eastern Pacific El Niños (Chen et al., 2016). Thus the anomaly and the large north-south gradient in the wind stress may be connected with the MJOs.

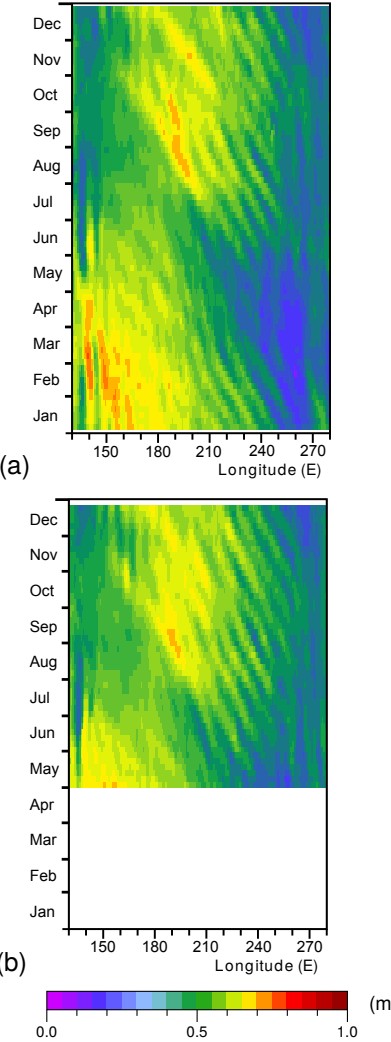

**Figure 9.** Hovmöller diagram of sea level (m) at 6°N during 1981, forced by the combination of 1981 and 1982 winds described in the main text when started from the NEMO archive for (a) 5th January 1981, (b) 30th April 1981.

Further east the ITCZ often lies close to 10°N. It is not clear whether this is related to the equatorial Rossby radius, but the figure shows that this is also a region where both the wind stress anomaly and its north-south gradient can be large.

To see if the local wind fields in the western Pacific were responsible for the drop in sea level in 1982, a modified wind field was constructed which combined both the 1981 and 1982 winds. The weighting for the 1982 winds was defined so that it equalled one within the region 140°E to 180°E and the Equator to 15°N and was zero outside the region 130°E to 190°W and 10°S to 25°N. Linear interpolation was used between the two boundaries. Weighting for the 1981 field was set to one minus the 1982 weighting.

**Table 1.** Mean sea levels (m) between 140°E and 170°E and between 5°N and 7°N, at the end of April and August for the different runs, identified by the model and start month. The two groups correspond to runs forced by the 1981 winds and by either the 1982 winds or the combined winds.

| | | | Start | Apr | Aug |
|---|---|---|---|---|---|
| 81 Winds | Occam | | Jan 81 | 0.620 | 0.557 |
| | ” | | Jan 82 | 0.632 | 0.581 |
| | ” | | Apr 81 | 0.633 | 0.520 |
| | Nemo | | Jan 81 | 0.605 | 0.570 |
| 82 Winds | Occam | | Jan 82 | 0.582 | 0.341 |
| | ” | | Jan 81 | 0.577 | 0.316 |
| | ” | | Apr 81 | 0.672 | 0.356 |
| | Nemo | | Jan 81 | 0.559 | 0.364 |
| Combined | Occam | | Jan 81 | 0.590 | 0.354 |
| | ” | | Apr 81 | 0.668 | 0.387 |

This results in the 1982 wind field being used for the area where the main sea level drop is observed and the 1981 wind fields being used elsewhere but with a smooth transition zone. The combined wind field was then used for two runs. As with the previous tests using 1982 winds, the first started from the ocean state for the 3rd January 1981 and the second from the 30th April 1981. If the drop in sea level is due to local winds then the change in sea level should be similar to those obtained using only the 1982 wind field.

The resulting sea levels along 6°N are shown in Fig. 9. In both cases sea level starts dropping in the western Pacific in early May, confirming the importance of the local winds after the 30th April.

## 6.1 Intercomparison

Fig. 10 shows the changes in sea level from the different runs, averaged over the region 140°E to 170°E and 5°N to 7°N, as outlined in Fig. 8. The Occam runs start with a slight offset from the corresponding runs of the Nemo model, due to the initial adjustment to the new coast and topography.

In this figure, colour is used to distinguish the different winds forcing the ocean, blue for 1981 and red for 1982. Even with the different starting conditions, the runs soon split into two distinct groups, sea levels with the 1981 winds staying roughly constant whereas the 1982 winds generate a 20 cm drop between March and August. As pressures in the deep ocean remain roughly constant, this drop in sea level must correspond to a significant rise in the density surfaces within the ocean.

To quantify the differences between the groups, sea levels from the end of April and August are given in Table 1, split into runs forced by 1981 winds and those forced either by 1982 winds or by the combined wind fields. A t-test shows that the

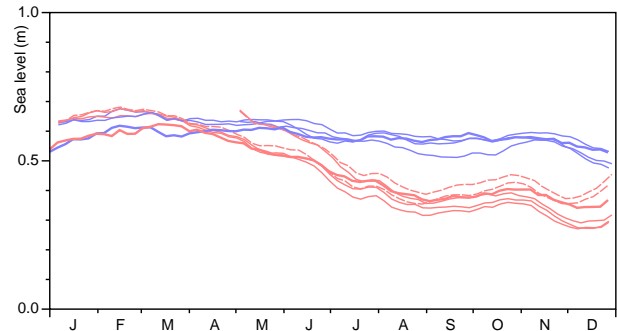

**Figure 10.** Sea level (m) in the different runs averaged between 140°E and 170°E and between 5°N and 7°N. Solid red lines correspond to runs forced by 1982 winds, blue to 1981 winds. Thick lines are from the original Nemo run, thin lines are from the Occam runs. The red dashed lines corresponds to the runs started in 1981 but with 1982 winds only in the western Pacific.

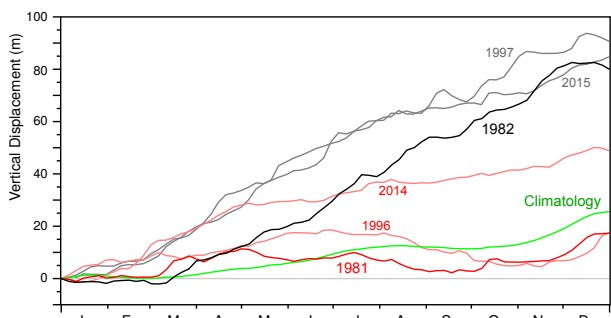

**Figure 11.** Integral over time of the vertical velocity due to Ekman divergence in the region bounded by 140°E, 170°E, 5°N and 7°N, during (black) 1982, (red) 1981, (green) climatology 1958-2015. In the background are the corresponding plots for the years prior to the 1997-1998 and 2015-2016 El Niños.

380 probability of both sets belonging to the same group at the end of April is around 0.5, as might be expected. By the end of August the probability drops to around 0.00002. A test using the sea level change over the period gives a similar value.

## 7  Winds and Ekman Divergence

Given that local winds appear to be responsible, the most likely cause of the rise in the density surfaces is that it is due to Ekman pumping, the result of a divergence in the wind generated Ekman transport in the surface layers of the ocean.

385   Here the hypothesis is investigated by calculating the Ekman pumping in the region 140°E to 170°E and 5°N to 7°N.

If the Ekman transport vector is $E(\theta, \phi)$, where $\theta$ and $\phi$ are latitude and longitude, then

$$E(\theta, \phi) \quad = \quad \tau(\theta, \phi) \wedge \hat{n}/(\rho f(\theta)) \tag{1}$$

where $\tau(\theta, \phi)$ is the wind stress vector, $\hat{n}$ the unit vertical vector, $\rho$ is the density and $f(\theta)$ is the Coriolis term, equal to,

$$f(\theta) \quad = \quad 2\,\Omega\,\sin(\theta). \tag{2}$$

$\Omega$ is the angular rotation rate of the Earth.

The averaged vertical velocity in a region, due the divergence of the Ekman flux, is then given by integrating the outward flowing Ekman transport around the region of interest,

$$P \quad = \quad -(1/A) \oint E(\theta; \phi) \wedge \hat{n} \cdot d\hat{s}, \tag{3}$$

$$= \quad (1/A) \oint \tau(\theta, \phi) \cdot d\hat{s}/(\rho f(\theta)). \tag{4}$$

where $\hat{s}$ is the unit vector tangential to the boundary and $A$ the area enclosed.

Fig. 11, shows the results obtained by integrating the vertical velocity over time. It shows that in 1982 the Ekman divergence (given by the slope of the curve) was positive for most of the period between April and late November, and that it had the potential for raising density surfaces within the ocean by 80 m. During the period mid-March to mid-August the potential rise is approximately 50 m.

A noticeable feature of Figs. 10 and 11 is that, during 1982, the drop in sea level and the vertical displacement due to the Ekman divergence are both relatively steady processes occurring over many months. There are short periods when the rate of change is reduced or reverses, but overall the results imply that the changes are a result of a long term systematic change in the wind field.

To check that the Ekman pumping is sufficient to cause the observed change in sea level, Fig. 12 shows density profiles from the original Nemo 1/12° model run averaged over the same area as before.

Figure 12a shows that during 1981, the changes in the depth of the density surfaces were small and at most depths had balanced out by the end of the year.

In contrast, during 1982 (Fig. 12b), there was a significant shallowing of the density surfaces. The figure shows that in mid-March, water with densities of 1025 kg m$^{-3}$ (approx 21.5°C) lay near 150 m, and that by mid-August it had risen to near 100 m. This rise of approximately 50 m is comparable with the Ekman pumping estimate. However after this period, although the negative Ekman pumping continued there was no further shallowing in the density surfaces near the surface and there is evidence of a rebound at depth.

## 8 The 1997-98 and 2015-16 El Niños

The results of the previous section support the hypothesis that Ekman pumping over many months was responsible for the drop of sea level at the start of the 1992-1993 El Niño. However if the mechanism is a significant part of all East Pacific El Niños then similar behaviour should occur prior to the 1997-1998 and 2015-2016 events. These are also periods when better wind field data was available so they should be better tests of the hypothesis.

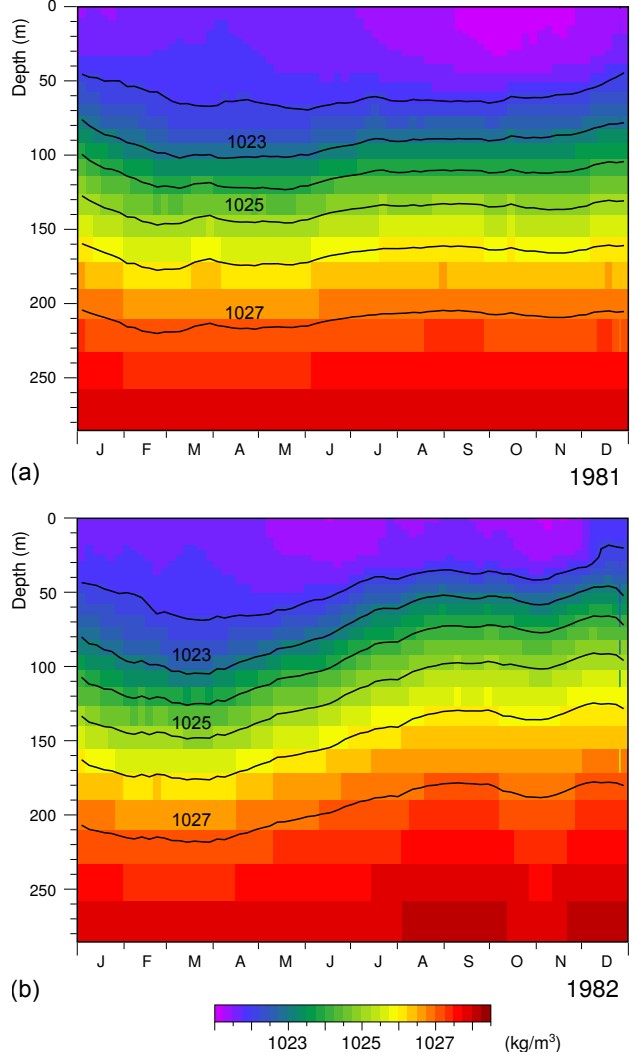

**Figure 12.** Nemo 1/12° model potential density averaged between 140°E and 170°E and between 5°N and 7°N, for (a) 1981 and (b) 1982. Contours at integer values of density $(\mathrm{kg\,m^{-3}})$.

The results, shown in Figs. 11 and 13, give support to the hypothesis, as in each case the vertical displacement is similar to the 1982 value. In both cases pumping continues almost continuously over many months, the only key difference being that in 420 both cases Ekman pumping starts earlier in the year.

Figure 11 also contains the results for 1996 and 2014. The results for 1996, like those for 1981 are similar to the climatology values. However in 2014 the displacement curve starts by following the curves for 1997 and 2015 and it is only after April that the pumping is reduced. The displacement then continues to increases until the end of the year.

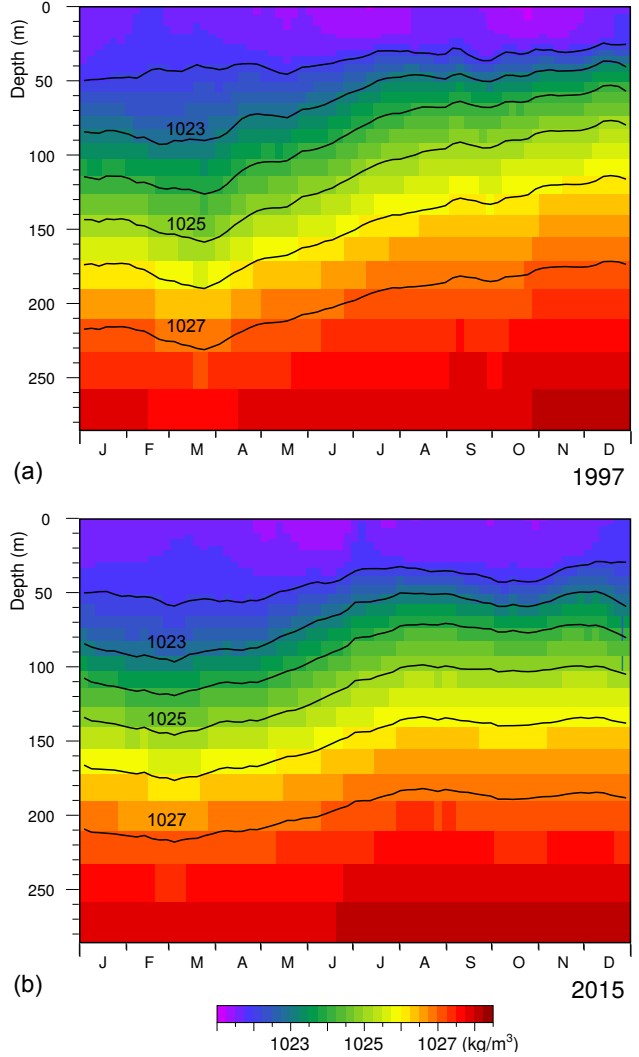

**Figure 13.** Nemo 1/12° model potential density averaged between 140°E and 170°E and between 5°N and 7°N, for (a) 1997 and (b) 2015. Contours at integer values of density $(\mathrm{kg\,m^{-3}})$.

A number of papers, including Min et al. (2015), Chiodi and Harrison (2017) and Wang and Hendon (2017), suggest that an eastern Pacific El Niño started to develop in 2014 but was then prevented by a change in the winds. It is possible that the 2014 curve in Fig. 11 reflects this event.

The main conclusion to be taken from the results is that, between mid-March and mid-August 1982, there was a large amount of Ekman pumping in the western Pacific around 6°N due to a wind anomaly. Similar amounts of Ekman pumping also occurred prior to the strong 1997-1998 and 2015-2016 El Niños. In each case the results indicate that the pumping was due to a systematic change in the local wind field which lasted for many months.

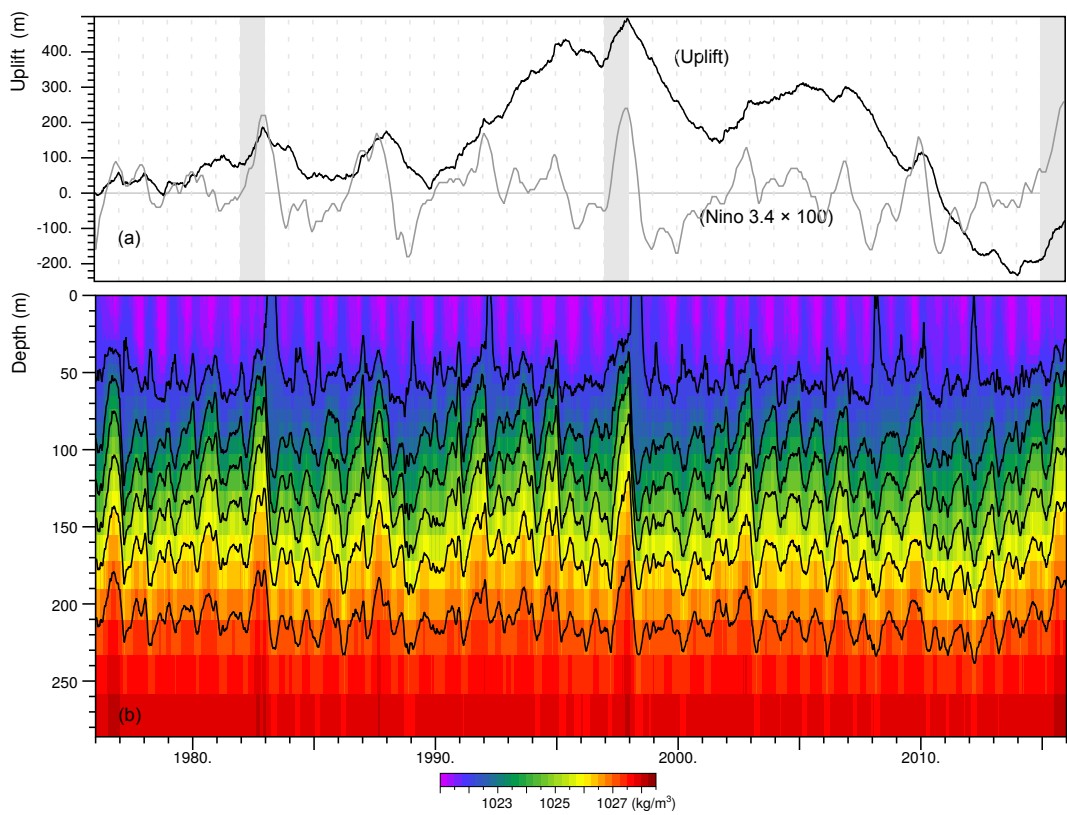

**Figure 14.** (a) Integrated Ekman pumping, based on wind observations, and (b) Nemo 1/12° model potential density, both averaged between 140°E and 170°E and between 5°N and 7°N, between 1976 and 2016. In (a) the grey line is the three month running average of the Nino 3.4 index (NOAA, 2021) multiplied by 100. The shading highlights 1982, 1997 and 2015 when strong eastern Pacific El Niños developed. In (b) the contours, interpolated from the density at model levels, are at integer values of density. The densest contour is at 1027 kg m$^{-3}$.

## 9 Developments between 1976 and 2016

By themselves the above results imply that strong Ekman pumping is always associated with strong El Niños and that in other years the integrated Ekman pumping remains small. However this is not correct.

Figure 14 plots the integrated Ekman pumping, the uplift, for the forty year period between 1976 and 2016. As before this is calculated from the Drakkar data set (Dussin et al., 2016) for the region between 140°E and 170°E and between 5°N and 7°N.

Also shown is the average potential density in the region from the Nemo 1/12° model and a scaled value of the Nino 3.4 index (NOAA, 2021) covering the same period.

The figure shows the significant amount of uplift that occurred prior to the strong El Niños of 1982-83, 1997-98 and 2015-16 and the corresponding rise in the ocean density surfaces. It also shows that similar amounts of uplift in 1997 and 2002 may be responsible for the shallowing of the density surfaces and the increased El Niño index.

Between 1990 and 1996 the figure shows a steady increase in uplift, with a mean slope only slightly lower than that seen prior to the strong EP El Niño years. However the El Niño index remains low, except for a period near the end of 1992 and a smaller event in 1995. Ekman pumping was also active early in the year in 1987 and 2002 but no strong EP El Niño developed.

The figure also shows that in late 2009, the El Niño index reached 1.6 and there was a corresponding shallowing of the density surfaces, but the estimated uplift is small.

Another feature of the figure is that as well as the period of positive Ekman uplift between 1990 and 1996, it shows significant drops in the integrated pumping between 1997 and 2002 and between 2007 and 2014. It is possible that the change in sign of the decadal average slope around 1996 is due to changes in the data available or the methods used by the ECMWF reanalysis project. However there is a small correlation between the sign of the El Niño index and the slope of the uplift curve, so the change before and after 1996 may be real.

Overall the results imply that, although Ekman pumping may be required to lower sea level in the North Equatorial Trough before a strong EP El Niño can develop, other factors are also likely to be involved. If this is correct it implies that Ekman pumping early in the year may be a necessary but not sufficient condition for the development of EP El Niños.

One obvious additional factor is that the volume of warm water available in the West Pacific Warm Pool needs to be sufficient. Although the differences in depths are small, this may help explain why the strong El Niño developed from years when the $1023 \ \mathrm{kg \, m^{-3}}$ density surface lies deeper in the ocean, as opposed to the 1990 to 1995 period when the average depth was shallower.

## 10   Conclusions

The first aim of this study was to understand the cause of the low sea levels that developed in the western Pacific, along the line of the North Equatorial Trough, during the growth of the strong 1982-1983 EP El Niño.

A comparison of results from the Nemo 1/12° and Occam 1/4° global ocean models indicated that the latter was suitable for studying the development of the 1982-1983 event. The Occam model was then used in a series of short tests, using starting dates in the same two years, to determine whether the state of the ocean or the wind field was responsible for the low sea levels.

The results showed that the key feature was the local wind field in the western equatorial Pacific and that, for these years, differences in the initial state of the ocean were not significant.

The most likely way that the local wind field can affect sea level is through Ekman pumping, which affects the ocean's stratification. The hypothesis, that this was the cause of the sea level drop in the North Equatorial Trough, was checked by using data from the original run of the 1/12° Nemo model to compare Ekman pumping in the western section of the trough with the changes in depth of the model's density surfaces.

The comparison showed that during the first few months of the year, the rise in density surfaces was close to that expected from Ekman pumping but that after mid-year, as the warm surface layer became thinner, the connection was broken, although Ekman pumping continued.

The comparison was then repeated for the period prior to the strong EP El Niños of 1997-98 and 2015-16, and similar results obtained.

As a further check on the hypothesis, the integrated Ekman pumping in the western region of the trough was calculated for the period 1976 to 2016 and compared with both the movement of the average density surfaces and the Nino 3.4 index during this period.

This showed that the presence of strong Ekman pumping during the first few months of the year was not sufficient by itself to trigger a strong EP El Nino. Ekman pumping may be a necessary condition but other factors, possibly including the volume

of water in the West Pacific Warm pool, must also be involved.

## 10.1 El Niños and westerly winds

Periods of positive El Niño index are often thought to be triggered by periods of westerly winds in the western Pacific associated with the Madden-Julian Oscillation.

One possible mechanism, connecting the two, requires westerly wind bursts to generate equatorial Kelvin waves which

eventually result in a warmer East Pacific Cold Pool. However there are doubts about this mechanism because the East Pacific Cold Pool region is never warm enough to trigger deep atmospheric convection.

However the westerly winds can also reverse the direction of the Equatorial Current in the western Pacific, and so transport Warm Pool water into the central Pacific. The temperature of the advected water is sufficient to trigger deep atmospheric convection and its effect on the position of the warm water front in the central Pacific may explain much of the normal year to

year variation in the El Niño index.

These mechanisms focus on waves and currents near the Equator but, as discussed earlier, observations show that water warm enough to cause deep atmospheric convection is carried even further eastwards by the North Equatorial Counter Current.

Webb (2018) investigated the cause of the strong 1982-83 and 1997-98 El Niños and argued they were due to the NECC advecting a larger than normal volume of Warm Pool water into the central and eastern Pacific as a result of low sea levels in

the North Equatorial Trough early in the year.

The study reported in this paper shows that these low sea levels are the result of local winds upwelling water into the surface layer. Although it is affected by the topography of the adjacent ocean continent, the northward extent of MJOs and other similar wind features in the western Pacific is close to the scale of the equatorial Rossby radius in the atmosphere and so includes the latitudes of the NECC.

As a result it is very possible that the mismatch between the equatorial Rossby radii in the ocean and atmosphere, means that MJO westerlies may trigger strong EP Pacific El Niños as a result of Ekman driven upwelling, at the latitude of the North Equatorial Trough, in addition to their impact on currents near the Equator.

*Code and data availability.* At the time of publication the Nemo model datasets are freely available at "http://gws-access.ceda.ac.uk/public/nemo/runs/ORC N06/means/". The Nemo ocean model code and its documentation are available from "http://forge.ipsl.jussieu.fr/nemo/wiki/Users". The

Occam model is based on the Moma ocean model available from "https://github.com/djwebb/moma".

*Author contributions.* The author is responsible both the model runs and the analysis.

*Acknowledgements.* This work contributed to and was aided by the research programme of the Marine Systems Modelling group at the UK National Oceanography Centre (NOC), part of the Natural Environment Research Council (NERC). NERC helped fund the investigation through the NOC National Capability funding. Part of the analysis was carried out using the JASMIN Service at the UK Centre for

Environmental Data Analysis, also funded by NERC.

The author also wishes to acknowledge the helpful comments and suggestions from Harry Bryden, D. Wang and three anonymous reviewers.

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
