# Peer review of "On the low western Pacific sea levels observed prior to strong East Pacific El Niños"

_Ocean Science, 2021_

## Community Comment (CC1)

Comments on "On the Role of Westerly Wind Anomalies in the Development of the 1982-1983 El Niño" by Webb

According to a series of model simulation experiments, the present study investigated the causes of the low sea levels that developed in the western Pacific around North Equatorial Trough in the development of 1982-1983 El Niño. The author suggested that the low sea level was due to the increased wind shear that developed just north of the Equator during 1982. Generally, the manuscript is more like a report rather than a research paper. There are too many figures which some of them should be merged to compare easily. Many plots of model experiments are shown, while physical progresses or mechanisms are less discussed. Some major comments are in the following.

Major comments

1. Several references about the influences of westerlies on El Nino are missed, i.e., Rasmusson and Carpenter (1982), McPhaden (1999), Li et al.(2005); Wang et al.(2011) and et al.

Li C, Pei S, Pu Y (2005) Dynamical impact of anomalous East-Asian winter monsoon on zonal wind over the equatorial western Pacific. Chin Sci Bull 50:1520–1526

McPhaden MJ (1999) Genesis and evolution of the 1997–1998 El Nino. Science 283:950–954

Rasmusson EM, Carpenter TH (1982) Variations in tropical sea surface temperature and surface wind fields associated with the Southern Oscillation/El Nino. Mon Weather Rev 110:354–384

Wang, X., C. Wang, W. Zhou, D. Wang, and J. Song, 2011: Teleconnected influence of North Atlantic sea surface temperature on the El Niño onset. Clim. Dyn., 37, 663-676

2. In Section 3, the author should give the observed sea level changes to validate the model simulations. In addition, this part including texts and figures are too much. About half of figures are used in Section 3, which is not necessary. It is suggested

to briefly introduce that the application of the Occam is suitable for the study.

3. Figures 13 and 16: Which figure is the "top" figure?

4. It is said that the anomalous westerly winds is associated with Madden Julian Oscillations, but there are any evidences. Pls confirm it.

5. The author showed so many plots of modeled SST, SSH and surface velocities and compared them. It is better to show the differences among them to make readers easily compare them.

---

## Author Response (AR1)

**Reviewer 1**

1.1 *This study investigates the origin of western equatorial Pacific sea level anomalies that seem to be related to the strong El Niño 1982/83 via modulating the North Equatorial Counter Current. This is motivated by a previous study about the importance of these sea level anomalies in driving the strong El Niño 1982/83. Using a global ocean general circulation model forced by different atmospheric wind fields, the author rules out different possibilities for causing the anomalously low sea level in the western equatorial Pacific, such as the annually occurring Rossby wave or a remote wind field, coming to the conclusion that it is the local wind anomaly field that drives the sea level anomalies driven by Ekman divergence.*

*The study is interesting as it sheds light on further processes that seem to be relevant in driving overly strong El Niño events beyond the conventional processes that are typically addressed in this context.*

No response

1.2 *The manuscript has rather the form of a report-type publication rather than a classical research article. However, this does not degrade the fact that it is informative. The methodology, the analytical reasoning and the writing are adequate.*

I do not understand the 'report-type publication' unless it means that a review is not included at the beginning of the paper. The introduction has been rewritten and expanded to cover this possibility.

1.3 *The overall quality of the current stage of the manuscript is however such that it is partly built too complicated, especially regarding the way the figures are presented (see specific comments).*

The figures have been combined to reduce the number of figures and to make their relationship clearer.

NOTE: Because of the figure changes, the text had to be reorganised. This is most noticeable in section 3 (Validation).

1.4 *There are some typos and some phrases that are difficult to understand (see technical comments)*

The revised manuscript includes a large number of minor improvements to the text.

1.5 *I suggest to rephrase the title of the manuscript to "On the origin of western equatorialPacific sea level anomalies prior to the 1982/83 El Niño"*

Change made.

1.6 *I strongly encourage the author to provide a few more references about driving mechanisms of the strong El Ninos and the role of anomalous wind fields in the introductory section, especially more recent ones.*

See 1.2 : New introduction.

*1.7   A lot of figures can be merged to reduce the total number of figures and also ease the comparison among them. For example, one could merge figures 1, 2, 7 and 8; figures 3, 4, 9 and 10; figures 5, 6, 11 and 12; figures 13 and 16; figures 14, 15, 17 and 18; figures 19 and 20; figures 25 and 26. The author may find a more reasonable order of merging some figure, but I think some merging should be done in a way that it reduces the total amount of figures and such that it puts those figures side-by-side that shouldbe compared to one another.*

See 1.3 : Revised figures

*1.8   The author may add the observed sea level and temperature field for that time period to the supplement to complement the model validation.*

Suitable data for 1981 and 1982 is not available. Webb et al (2020) showed that the Nemo model successfully reproduced the SSH and SST fields in the period 1995-2000.  The paper assumes that it did equally well for the period 1980-1985.

*1.9   There are some typos such that double-occuring words: "from from", "that that" –please search for these occurrences and correct.*

See 1.4: Many minor errors corrected.

1.10  *I also suggest to add the information of the physical units to the figure captions.*

Units added.

1.10  *I also think that is called "Hovmoeller" rather than "Hovmuller" diagrams.*

Spelling corrected.

*1.11  147-148: Please check structure and meaning of the text - unclear*

Section revised..

**Reviewer 02**

*2.1   The role of observational uncertainty has not been but also should be considered in this context. Previously, for example, Harrison et al. (1990) reported on forced ocean model hindcasts of the 1982-83 El Nino event and found that the answers to questions like those being asked presently, for example, concerning the relative importance of local and remote wind forcing to anomalous currents and SST, depended very much on which wind data set was used to force the model. The present manuscript appears to report results based on only one wind data set, which is not described in the text. Given the previous Harrison et al. demonstration of the importance/limitations of observational wind uncertainty in this context, the impact of this wind uncertainty needs to be examined before the reliability of the results presented can be understood.*

The revised paper contains text designed to answer this point.  The introduction refers to the poor representation of the NECC in many ocean-only models - which Yu et al (2000) showed was due to problems in representing the curl of the wind stress near the latitude of the ITCZ

and overestimating the wind stress on the Equator. Their work showed that the ECMWF reanalysis had the smallest error.

The Nemo model (and the Occam model here) used the Drakkar dataset which is based on the ECMWF with improvements in the equatorial Pacific (see the Drakkar reference for details). The agreement between Nemo and observations reported in Webb et al (2020) indicate that any remaining error in the wind stresses must be small.

The paper reports on the good behaviour of the ECMWF analysis, the further improvement of the Drakkar dataset and the and the good agreement with observations of the Nemo results. Other datasets may have problems but I hope this is enough to cover the forcing used here.

*2.2  Notwithstanding the issues raised in the comments above, more precise description of the experiment results would improve their presentation (and facilitate comparison to observations). This manuscript relies mainly on visual inspection of snapshot-maps and time-longitude plots of SST, SSH and currents to support its conclusions about the relative importance of different ocean initial conditions and components of wind variability for causing changes in NECC-related SSH. I suggest defining metrics that quantify the salient model experiment results in relation to the control-hindcast to thereby offer a more streamlined and precise presentation of results.*

I am not sure what the reviewer had in mind but to this end in section 6.1 I report on a t-test comparing the response of the different runs to 1981 and 1982 winds. The result is fairly conclusive.

2.3  *I suggest that "Westerly Wind Events" be removed from the title. This manuscript does not identify or directly discuss westerly wind events. Something like "On the development of low North Equatorial Pacific sea level pressure during 1982-1983" would be more appropriate.*

See 1.5

*2.4  The last paragraph of the abstract attempts to describe the relationship between westerly wind events, the Madden Julian Oscillation, North Equatorial Counter Current (NECC) and the observed development of the 1982-83 El Nino event based on the NECC-related model experiments presented herein. However, what is presented herein does not sufficiently support conclusions about these relationships because three of these four phenomena (wind events, MJO, observed El Nino development) are not substantially addressed by the results presented in the manuscript. The abstract should be modified to better reflect what has and has not been done here.*

I do not really understand the reviewers problem here and take issue with what they are expecting. The abstract does contain the provision "If Webb (2018) is correct .." and given the results of that paper the rest of the paragraph follows.

2.5  *Many of the model comparison figures, for example Figs. 3&4, 5&6 etc. can be combined to the benefit of the reader's ability to make the intended visual comparison. Reducing the total number of figures may also improve the presentation; 24 is perhaps an over- abundance of figures for the scope of this paper.*

See 1.3 : Revised figures.

2.6    *There has been considerable progress made in understanding El Nino development since Wyrtki (1973, 1974) offered hypotheses about the role of enhanced NECC. This manuscript would benefit from taking into account what has been learned and described in*

See 1.2 : Revised introduction

2.7    *Specific Comments*

The paper contains a large number of minor improvements including those referred to here.

\*    *Paragraph beginning Line 117 and associated Figures. The key, near-equatorial features are difficult to see with latitudes +/- 30 and surface current vectors shown. The author may wish to consider reducing the Y-axis range to facilitate visual inspection of the most import model results.*

On the question of latitude range, the use of 30S to 30N arises partly from the requirement to include the whole of the longitude range of the Pacific without much distortion.  It has the advantage of including the tropics and sub tropics.

Once on-line the paper will most often be read using a pdf viewer.  In the original manuscript the individual figures contained the underlying postscript at full resolution allowing readers to magnify the figures to see every detail.

Unfortunately I could not find software which could combine the figures at full resolution without entering some infinite loop.  Instead for the revised manuscript the longitude-latitude figures use a high resolution jpeg images.  These are not so good but still allow a large amount of magnification.

\*    *Paragraph beginning on Line 252. Integration smooths any field regardless of whether its variability is characterized by a "long term systematic change", or a more event-like abundance of, for example, equatorial westerly anomalies. Results therefore may not imply what the text claims they do*

I would agree that the results do not 'prove' that a systematic change in the wind field is involved but I think that the word 'imply' is valid - especially given how poorly MJOs and similar events are understood.

\*    *Line 300. The observed variability of coupled tropical Pacific system does not support a one-to-one correspondence between equatorial ocean current variability and the observed variables most closely related to deep atmospheric convection activity, such as is implied here. The statement about convection should be sufficiently supported or withdrawn.*

Text changed to "would have extended the region over which deep atmospheric convection could occur."

**Reviewer 3**

3.1    *26 figures for a paper with a relatively short story seem way too many. E.g., there is no need for 16 figures only for model validation. Why not validate the model with one or two timeseries averaged over selected regions and only show one map for one date and one year as an example?*

*In general, throughout the manuscript it would be very helpful for the reader to have all related panels closer together (e.g., in one figure) so they can be seen and compared at once without flipping pages. Also, difference plots would be very helpful for the model validation as well as in the results section. A lot of the figures can be merged together to one figure with more panels.*

*Another comment regarding the appearance of the figures. It should be made sure that all figures/panels are consistent among each other (e.g., same axis labels). Also, many (all?) axis labels and axis tick labels are very tiny and could be made larger.*

The figures have been reorganised and reduced in number to 12 with key figures close together and with increased typeface size for many of the labels. The validation might have been done in a different way but there then could be arguments about whether the regions selected were sufficient.

I have not added the difference plots as (a) It would have involved more plots and (b) I do not see how they would have contributed to the final results.

3.2 *Why is the author only focusing on the 1982/83 event and not also the 1997/98 event? It would be very interesting to know if the presented mechanism also applies to other strong El Nino events.*

The original reason was that this is the event which I had studied most. I've added the curves for 1996, 1997, 2014 and 2015 to Fig. 11 of the revised manuscript and commented on the differences.

3.3 *Section 1.2 seems unnecessarily long as it already goes into the details of the methods and even results. It could be condensed by leaving out all the details (shifting them to the Methods section) that are partly mentioned in the following sections and merged to the end of section 1.1.*

For many readers skimming through the paper, I think this is worthwhile as it outlines the main arguments of the paper and allows them to focus on the aspects that they think are most important.

3.4 *There are multiple small spelling and grammatical mistakes (mostly missing words) throughout the manuscript. The manuscript should be carefully checked against such mistakes as it makes it harder for the reader to follow. I have started with a few in the specific comments below but they became too many to all list them here.*

Typos have been corrected. The manuscript also contains many other small improvements.

3.5 *Specific Comments*

* *line 113: Why specifically these dates, 4th June and 2nd September? Probably related to the build-up and major phase of NECC transport. The author should explain/justify this.*

Explanation added.

\*       *Figures 3-6: In general, Occam seems to overestimate sea level variations as compared to Nemo. This should be mentioned in the discussion of the Figures. What are possible reasons for this? What impact on the results does this have?*

*lines 133-136: It should also be concluded that due to the lack of heat and freshwater fluxes Occam does not well capture the SST features (amplitude) of Nemo? How does this affect the usefulness of the model for this study?*

With the re-organisation of the figures, the text comparing Occam and Nemo has also been revised.  I include a lot on the temperature problem as well as sea level but in section 3.3 conclude with what I think is a fair summary:

"However by bearing these strengths and weaknesses into account, there is no reason why it cannot be used to study the effect of the winds and the initial state of the ocean on the development of the 1982-1983 El Niño, as is done here."

Of course in the end the results from Occam are only used to provide hints as to why sea level dropped in the Nemo runs - the relationship between the wind curl in the forcing dataset and the sea level drop in Nemo being the key result of the paper.

*lines 133-136*
This was a typo error.

\*       *lines 198-203: Is this experiment necessary, after all, as the author has already shown in Figure 13 that the 1981 winds cannot produce a sea level low in the western Pacific?*

See text: "One possibility that has not been discounted is that winds early in the 1982 generated a Rossby wave, or similar, which was later responsible for the drop sea level drop in the western Pacific."

\*       *213-214: It is unclear to me how this statement contributes to the overall story presented in the manuscript. Please clarify.*

First, although I do not go into detail within the paper, an observant reader might notice that the wind field anomaly changes between the far western Pacific, where westerly wind bursts and the MJO are most likely involved, and more central longitudes, where the ITCZ is most likely to be dominant.

Secondly it reminds the reader that both the scale of MJOs and the latitude of the ITCZ - both poorly understood atmospheric features - are close to the atmospheric equatorial Rossby radius.  In any decent physics course natural scales would be one of the first topics discussed.

\*       *Figure 22: Is panel a (left) really necessary as it was already shown before that the winds before 30th April do not play a role here.*

This is one of the key results of the paper.  A second run starting from an independent initial condition provides some additional support to the result.

\*       *Figures 25-26: Why is the Nemo output shown here instead of the Occam output?*

Nemo was calibrated against observations in Webb et al 2020.  Nemo was forced by a full surface flux data set.  Nemo used a much higher resolution,

*line 256: I would say the Ekman divergence causes sea level changes, rather than Ekman pumping which causes isopycnal changes.*

It is a moot point, divergence lowers sea level by a small amount, which generates a baroclinic response, pumping, which changes the density field, which determines sea level once the pumping has stopped. My focus is on the final sea level.

*lines 262-264: What is the reason for this? Is it because of an increase in temperature as the El Nino develops?*

I suspect it is because of increased stratification near the surface but Rossby waves may be involved - anyway I, thought that this was something for another time.

---

## Referee Report (RR1)

Review #2 of "On the Role of Westerly Wind Anomalies in the Development of the 1982-1983 El Niño", for the Journal of Ocean Science.

The author has adequately responded to my comments by drastically reducing the number of figures, by clarifying open questions and by adding text passages at respective locations in the manuscript. Thus, I can now suggest publication.

Only one comment has not been responded to, although I would be very curious about the author's opinion (my comment on lines 184-187).

Specific comments:

lines 74-75: Shouldn't this say "anomalous SST" instead of "mean SST"?

---

## Author Response (AR2)

```
05_Comments_01
==============
```

    Summary

    Reviewer 1  :  Excellent : Good : Good : Accepted as is
    Reviewer 2  :  Poor      : Poor : Fair : Rejected
    Reviewer 3  :  Good      : Fair : Fair : Minor revisions

    I would like to thank all reviewers for taking time on this paper.  I note
the 'noise' (Lawton, New Scientist, 19 June 2021, p40-44).

    I note that the editor also asked for a major revision.

```
Major changes
=============
```

**1 Changes to title and text**

One of the main problems comes from the title, which focuses on the 1982-83 El Nino.  It is true that the first section of the paper concentrates on this period but this is only to try and understand why the Nemo model generated reduced sea levels at the western end of the North Equatorial Trough in that period.  There was no intention to say the Occam model was perfect - just that it reproduced the Nemo results and so, although faster to run, could be used to study why the sea level change occurred.

Once a hypothesis was developed, this was then tested against the observed data, i.e. the reanalysed wind fields, prior to the east Pacific El Ninos of 1997-98 and 2015-16.

What this means that it is not essential for the occam model to be validated against observational data in the early 1980s only that it is validated against the Nemo model.

In the present set of reviews, the focus on the 1980s underlies many of the reviewers criticisms and especially their request for validation of the occam model during this period.  I have therefore changed the title and rewritten parts of the paper to emphasise the split between the hypothesis development phase reported in the first part of the paper, where validation against the Nemo model was required, and the testing/validation phase reported in the second part of the paper.

**2 Extra figures**

To emphasise this point I have also added two figures and their associated text.  This increase partly reflects the much greater importance of the second part of the paper.

The first figure illustrates the rise in the density surfaces within the Nemo model during 1997 and 2015, to show the effect of Ekman pumping in these years.  The second includes the Nemo forcing estimates of sea level change and the resulting changes in the Nemo density surfaces from 1976 onwards, to put in context the changes that occurred prior to the strong El Ninos.

```
Detailed comments
=================
```

1.  Review 1

    Many thanks.

2.  Review 2

    2.1  The reviewer is concerned that the paper does not contain a validation against observational data from the period 1981-1983. Reviewer 3 makes a similar point.

    In response:

    First, the initial part of the paper is concerned with understanding why the Nemo model generated low sea level at the western end of the North Equatorial Trough prior the 1982/83 El Nino.  If sufficient computer time had been available the Nemo model itself would have been used for these investigation but here section 3 is used to check that the Occam model is behaving similarly to the Nemo model in key regions during this period and is not affected by its lower resolution (see comments on sub-grid processes below).

    Secondly there is a general problem in validating models such as Nemo, Occam or any of the models used in climate research, in that they are so complex that there is no way that they could be properly validated as part of a normal length paper.  There is also no guarantee that a validated model does not misrepresent some other key aspect which affects the problem being tackled.

    Instead four tests are usually used.  The first is that the model is known to be based on good physics.  The second is that it has been successfully used in the past without errors which might affect the present study.  The third, a trivial one, is that the results are sensible given the known properties of the ocean.  The fourth, which is less common, is that the model can make predictions which can be successfully compared against observations.

    Concerning the first two tests, both Nemo and Occam have been developed by expert modelling groups and are well documented.  They have both also been used successfully in many studies of the ocean in many regions of the world ocean.  All ocean (and atmospheric) models have problems with sub-grid processes but here there is no evidence that they are a significant problem.

    In Webb (2018), the analysis of Nemo during the development of the strong 1982/83 El Nino was used to predict events that would occur during the development of the strong 1997/98 El Nino.  The same paper showed that these events did occur in the Nemo model run itself during the later period. The study of Webb, Coward and Snaith (2020) also confirmed that they occurred in the much better observational data from the later period.

    In the present paper the initial study of the 1982/83 El Nino generates the hypothesis that Ekman pumping was responsible for the initial drop of sea level in the North Equatorial Trough.  The resulting prediction is that similar strong Ekman pumping over many months would be observed prior to the strong El Ninos of 1987/1998 and 2015/2016.

    The prediction did not require further runs of the Occam model as the prediction could be tested directly against the wind stress fields.  Because of the increased availability of good scatterometer data, these would would have been much more accurate at the times of the 1997/1998 and

2015/2016 El Ninos than for the 1982/1983 El Nino. The results presented confirm the prediction.

Because of the above points, it appears that there is nothing to be gained by adding a comparison with 1982/1983 observations - although they may be useful in a separate paper which tests the mechanisms discussed in Webb (2018).

2.2 The reviewer believes that I did not take into account his previous comment that "model answers very much depended on which wind data set was used to estimate the surface forcing".

In fact I did take his comments into account and this is the reason why the introduction section contains so much on the wind datasets, their problems and the (possibly lucky) choice of using the ECMWF reanalysis. I thought I had included a reference to the Harrison paper itself but see that I didn't. This omission is now corrected.

2.3 "In my estimation ...".

I do not doubt the reviewer's belief but I hope that after considering the above arguments they will agree that hypothesis development followed by testing of predictions against observational data is equally (or even more) valid.

**3. Reviewer 3**

3.1 I accept that other people would work in different ways. I did not use climatology because this would have included the contribution from La Nina years. The only other option I considered was the use an average of years with a small El Nino index.

The advantage I saw for using the 1981 ocean was that the background stratification was as close as possible to that at the start of 1982. Similarly the 1981 winds, which must in some way reflect the global 1981 ocean, should be close to what the 1982 winds would have been if the El Nino had not started developing.

If the Ekman pumping hypotheses had been found later not to fit the Nemo model results or the 1997/98 observations, then I agree that the winds would need to be investigated in more detail. But once the hypothesis was developed and shown to be not unreasonable this was not needed.

3.2 I agree that the initial state of the ocean is important - obviously the mechanism proposed would not be effective immediately after a strong El Nino when the pool of warm water in the western Pacific is much reduced.

For this reason I have changed the wording referred to.

The next part of the reviewer's comment refers to the ocean's initial condition, discussed above. The remainder is then primarily concerned with the role of westerly wind bursts in different years.

The present paper does discuss the differences in the development of the 1997/98 and 2015/16 El Ninos and includes references to the unusual behaviour in 2014.

On the question of westerly wind bursts and the additional references given by the reviewer I have two problems.  First, westerly wind bursts do not really affect the conclusions of the present paper.  Secondly they represent an alternative mechanism for the development of strong El Ninos.

In order to properly discuss westerly wind bursts, the manuscript would need an additional section which argues for and against the two mechanisms.  Having thought about it I think this is best done by other people who from a distance can dispassionately judge the competing evidence and arguments.

3.3  Comparison with observations.  As discussed in the response to reviewer 2, the test of the prediction of enhanced Ekman suction at the start of the 1997-1998 and 2015-2016 El Ninos - using better wind stress data available for those periods - is a comparison with observations.  In both cases the prediction was successful.

I thank the reviewer for the references for data from the 1982/83.  I have used these in the revised paper.

3.4  My focus on 1982 is covered in my response to reviewer 2.

3.5  Equatorial currents.  I agree that changes in the direction of the Equatorial Current in the west Pacific due to the winds is a key part of the normal El Nino/La Nina process.  This has the effect of changing the eastward extent of warm water on the equator and so can significantly change deep atmospheric convection and the large scale atmospheric circulation.

However one of the points of the Webb (2018) paper is that during the development of strong El Ninos warm water transported by the NECC carries warm water much further eastwards, so causing an east Pacific El Nino as opposed to a central Pacific El Nino.

In the reviewed version of the manuscript I do mention the role of the Equatorial Current in transporting warm water eastwards.  In the revised version the text has been expanded.

Manuscript Revisions
====================

Reviewers 2 and 3 are primarily concerned about the use of the 1982/1983 in the first part of this study, the lack of comparison with observations during this period and the lack of discussion of other mechanisms - especially the accepted theories surrounding the role of westerly wind bursts and currents in the equatorial wave guide.

My (honest) response has been to provide arguments of why major changes to the paper to address these points are not appropriate.  However they are intelligent people - so I suspect that the real problem is background, i.e things I take for granted given my background compared with things they take for granted or are concerned about.

As detailed in "Changes to title and text" I have therefore skipped many of the detailed requests and instead have made changes which, I hope, clarifies my argument and, I also hope, means the paper is better suited for publication.  I would like to thank the reviewers again for their time and effort (and for the typos) and hope that they agree the final result is worthwhile.

---

## Author Response (AR3)

Authors Message to Production
==============================

The first four figures are pdf files of 4 MB and 2.9 MB. The rest are all less than 600 KB.

Please allow the forst four figures to be available as high-resoution figures from the on-line version of the paper.

David Webb